# Social and Health Beliefs Related to College Students’ COVID-19 Preventive Behavior

**DOI:** 10.3390/healthcare11131869

**Published:** 2023-06-27

**Authors:** Nam-Yi Kim

**Affiliations:** College of Nursing, Konyang University, Daejeon 35365, Republic of Korea; namyi00@konyang.ac.kr; Tel.: +82-42-600-8586

**Keywords:** COVID-19, health behavior, health beliefs, perception, social beliefs, students

## Abstract

Coronavirus disease 2019 (COVID-19) infection prevention behaviors vary from individual to individual, and it is necessary to identify factors related to such behaviors. This study investigated how college students’ social beliefs and health beliefs are related to their adherence to COVID-19 precautionary behaviors. An online survey was conducted among 200 Korean college students from 4 March to 30 June 2022. The variables associated with COVID-19 precautionary behaviors were evaluated, with social beliefs as the independent variable, health beliefs as the mediating variable, and COVID-19 precautionary behaviors as the dependent variable. A correlation analysis and confirmatory factor analysis were performed. The model fit was as follows: χ^2^/degrees of freedom = 1.64 (*p* < 0.001), Tucker–Lewis Index = 0.92, comparative fit index = 0.93, standardized root mean square residual = 0.06, and root mean square error of approximation = 0.06. Social complexity, as perceived by college students, was related to COVID-19 precautionary behaviors through mediating health beliefs (perceived benefits). To increase college students’ compliance with COVID-19 precautionary behaviors, it is necessary to identify social beliefs and accordingly propose interventions that focus on personal health beliefs.

## 1. Introduction

During the public health crisis caused by coronavirus disease 2019 (COVID-19), various influences, including demographic characteristics (e.g., age), the availability of ventilators, and government policies, led to differing mortality and incidence rates for COVID-19 [1]. It has become increasingly clear that the health care choices that were made, especially in the early years of the pandemic, had an impact on the course of the pandemic [2]. The size and complexity of the COVID-19 case population have led to additional unreasonable and inconsistent management of infected cases [1,2], making it important to devise a method of providing reasonably consistent guidelines in the event of an additional infectious disease outbreak in the future. In the Republic of Korea, the first COVID-19 case was confirmed on 20 January 2020, and as the infection reproduction index increased from 1.21 to 1.29, the four-step policy of distancing began on 12 July 2021 [3]. In the fourth stage of COVID-19 distancing, universities across the country switched to completely remote classes, and restrictions on gatherings of more than four people were implemented [3]. Social restrictions were partially effective in inhibiting the spread of COVID-19 [4], although there were individual differences in adherence to infection prevention behaviors [5]. In particular, variants that are unsusceptible to the existing vaccines have continued to emerge, necessitating personal preventive behaviors to reduce the infection spread [6]. Therefore, an individual-level examination of COVID-19 precautionary behaviors is required for developing informed measures to improve personal infection prevention behaviors.

College students, who are in the early stages of adulthood, potentially have greater COVID-19 exposure through various activities as they become increasingly socially independent [7], as well as exposure to various lifestyle-related issues, such as smoking, drinking, and irregular dietary habits [8]. According to the US Centers for Disease Control and Prevention, about 17% of people infected with COVID-19 are between the ages of 18 and 29 [9]. In Korea, 15% of infected people are between the ages of 20 and 29, which is a higher rate than among children and adolescents or older adults [10]. Moreover, young adults have little interest in diseases and tend to neglect health awareness and health promotion behaviors [11]. Furthermore, the younger population is more likely to have asymptomatic COVID-19 infection than the older population [7]. Consequently, asymptomatic young adults can transmit the infection to others who have higher risks. Therefore, following infection prevention behaviors during early adulthood and examining the factors that affect the behaviors of young people are necessary.

Individual health beliefs can be applied to explain and predict infection prevention behaviors because they focus on various factors, such as disease-related fear and anxiety. The health belief model (HBM) is a value-expectancy theory that refers to the desire to avoid illness and the belief that health-related behaviors will prevent illness [12]. Among the HBM components, the major variables for inducing health behaviors include the perceived susceptibility (belief about the risk of an illness), perceived severity (belief about the severity of the outcomes of illness), perceived benefits (belief that practicing preventive behaviors can reduce the probability of acquiring a specific illness and the negative consequences of the illness), and perceived barriers (belief about how difficult it is to overcome economic and psychological barriers for preventive behaviors) [12,13]. The HBM has been successfully applied to strengthen preventive behaviors for human immunodeficiency virus [14], respiratory infection [15], and nosocomial infections [16]. Therefore, we attempted to confirm the relationship with COVID-19 infection prevention behaviors by applying major variables in the HBM.

Social beliefs constitute a factor that can be applied to explain and predict infection prevention behaviors at the social level [12]. Leung et al. [17] identified that universal and general social beliefs about individuals and the social and physical environments influenced individual behaviors [17]. Social beliefs include social cynicism (negative view of human nature, prejudice against social groups, and distrust of social institutions), a reward for application (belief that people’s use of effort, knowledge, careful planning, and other resources will lead to positive outcomes), social complexity (belief that people’s behavior may vary across situations and that problems have multiple solutions), fate control (belief that life events are predetermined by various external forces, but that there are ways for people to predict and change their destiny), and religiosity (belief in the existence of a supernatural being and that religious practice has a beneficial function) [12,17]. These social beliefs provide guidance on human behaviors, including health and safety behaviors, and explain behaviors related to perceived causes and interactions [17]. Therefore, they could be utilized to explain COVID-19 precautionary behaviors.

Previous studies on COVID-19 precautionary behaviors investigated the knowledge and educational needs [18,19], risk perception [20,21], information-seeking tendency and health literacy [22,23], and attitude [24] of college students. Kim et al. [18] confirmed statistically significant positive correlations between college students’ knowledge of COVID-19, educational needs, and preventive behaviors. In a study by Taghrir et al. [21], COVID-19 infection prevention behavior and risk perception showed a negative correlation. Maheshwari et al. [24] suggested that knowledge about COVID-19 and positive attitudes toward infection prevention behavior should be improved. However, few studies have been conducted on college students’ social and personal health beliefs. A multidimensional perspective on social beliefs and individual health beliefs is needed to explain and predict individual behavior.

Social beliefs are appropriate for explaining different types of human beliefs, perceptions, and behaviors in different cultures [25,26]. Previous human immunodeficiency virus-related studies have also confirmed that social beliefs influence individual health beliefs and behaviors based on the social action theory [27]. Therefore, this study assumed that college students’ social beliefs would act as a leading factor in individual health beliefs and affect COVID-19 infection prevention behavior. The purposes of this study were to (1) identify the correlation between social beliefs and health beliefs regarding college students’ COVID-19 infection prevention behaviors and (2) identify the mediating effect of health beliefs in the relationship between college students’ social beliefs and COVID-19 infection prevention behavior.

## 2. Materials and Methods

### 2.1. Design

This survey was designed to examine how the social and health beliefs of college students in the Republic of Korea affect their COVID-19 precautionary behaviors. The independent variables were social cynicism and social complexity; the mediating variables were the perceived severity, perceived barriers, perceived susceptibility, and perceived benefits; and COVID-19 precautionary behavior was the dependent variable. A structural equation model was applied to simultaneously estimate the individual but interdependent relationships of the hypothesized model and to confirm its goodness-of-fit. Figure 1 shows the hypothetical pathways of this study.

### 2.2. Participants and Data Collection

For this study, an online survey was conducted among college students in the Republic of Korea between 4 March and 30 June 2022. Research participants were recruited using a Korean site (www.everytime.kr accessed on 4 March 2022; university student-only community) to which college students are subscribed. Any college student could freely participate in any field of study. In order to rule out the consequences of differences in academic backgrounds, no specific schools or majors (e.g., nursing, medicine, health) were presented. The inclusion criteria were (1) those who were currently enrolled as university students in Korea and (2) those who read and agreed with the study description. The exclusion criterion was those who were on a leave of absence from college at the time of this study. As this is a structural equation study, the sample size needed to be 10 to 20 times the number of variables [28]. The maximum number of variables was expected to be 14; therefore, a sample size of 140 to 280 was required. However, the maximum likelihood method allows sample sizes of 150 to 200, if the assumption of normality is satisfied [28,29]. Accordingly, data from 200 college students were collected while considering potential dropouts. As no responses needed to be eliminated based on the inclusion and exclusion criteria and the criterion of missing values of ≥10%, data from all 200 participants were included in the final analysis.

Before study initiation, ethics approval was obtained from the Konyang University Institutional Review Board (approval number: KYU 2021-12-010-001). Voluntary participation was invited through a recruitment notice on an online bulletin board. The recruitment notice explained how the collected data would be managed, that personal information would be protected, and that participants had the right to withdraw from this study at any time. The notice also included the link to the online survey.

### 2.3. Research Instruments

#### 2.3.1. Social Beliefs

Social beliefs were assessed through social cynicism and social complexity factors in the tools presented by Leung et al. [17], who described the five elements of social cynicism, a reward for application, complexity, fate control, and religion, as factors of social beliefs and mentioned that each element could be used independently. This study focused on identifying how perceptions of the social and physical environment affect individual beliefs and behaviors. Therefore, variables related to supernatural powers or religious beliefs (fate control, religion) were excluded. A reward for application is a variable related to individual perception (perceived benefit) rather than perception of social and physical environments, and it was excluded to avoid duplication with individual health beliefs presented as parameters. The content validity was confirmed by experts (three nursing professors and three infection control experts) to adopt Lung’s social cynicism and social complexity measurement instruments for college students. All items had a content validity index of ≥0.80; thus, the instruments were used without modification. Social cynicism measures the degree to which respondents believe that human nature and the social world will bring negative outcomes (e.g., “People create barriers to hinder the success of others”), whereas social complexity measures beliefs about the complexity and variability of the world (e.g., “People may behave in opposite ways on different occasions”). The questionnaire consisted of 16 items, with 8 items each for social cynicism and social complexity. Each item was rated on a 5-point Likert scale (1 = strongly disagree to 5 = strongly agree), with higher scores indicating stronger social beliefs. The reliability of the instrument (Cronbach’s α) for social cynicism and social complexity was 0.79 and 0.74, respectively, in Leung et al.’s study [17], and 0.81 and 0.87, respectively, in this study.

#### 2.3.2. Personal Beliefs

The perceived benefits, perceived severity, perceived susceptibility, and perceived barriers, which are major components of the HBM used in Kim and Jeong’s study [30], were used to measure personal beliefs. The term “bloodborne infection” from Kim and Jeong’s study [30] was changed to “COVID-19 infection” for this study. The instrument consisted of a total of 24 items, each rated on a 5-point Likert scale as follows: 5 items for perceived benefits (1 = not effective at all to 5 = very effective; e.g., “Do you think COVID-19 prevention measures are effective in maintaining your health?”); 5 items for the perceived severity (1 = not serious at all to 5 = very serious; e.g., “Suppose you contract COVID-19; what impact will it have on your social life?”); 5 items for the perceived susceptibility (1 = not at all to 5 = very much; e.g., “Has anyone around you had COVID-19?”); and 9 items for perceived barriers (1 = not burdensome at all to 5 = very burdensome; e.g., “Is the time spent engaging in COVID-19 infection prevention a burden?”). For the perceived benefits, perceived susceptibility, and perceived severity, higher scores indicated a higher perception of infection risk, whereas for perceived barriers, higher scores indicated a lower perception of infection risk. The reliability (Cronbach’s α) of the instrument was 0.79 in Kim and Jeong’s study [30] and 0.74 in this study. The reliability (Cronbach’s α) for the perceived benefits, perceived susceptibility, perceived severity, and perceived barriers in this study was 0.88, 0.83, 0.85, and 0.91, respectively.

#### 2.3.3. COVID-19 Precautionary Behaviors

To measure COVID-19 precautionary behaviors, the questionnaire was prepared using the national code of conduct defined by the Korea Disease Control and Prevention Agency (Ministry of Health and Welfare) and the infection prevention behavior scale developed by Kwak and Kim [31]. The questionnaire consisted of 10 items on mask wearing, hand hygiene, social distancing, and a symptom check (e.g., “I always wear a mask” and “I keep 2 m distance from other people”). Each item was rated on a 5-point Likert scale (1 = strongly disagree to 5 = strongly agree), with higher scores indicating a higher level of precautionary behaviors. The reliability (Cronbach’s α) of the instrument was 0.81 in Kwak and Kim’s study [31] and 0.85 in this study.

#### 2.3.4. Sociodemographic Variables

Five sociodemographic variables were considered: age, gender, grade, source of information related to COVID-19, and isolation experience related to COVID-19.

### 2.4. Data Analysis

Data were analyzed using SPSS (version 25.0; IBM Corp., Armonk, NY, USA) and AMOS (version 28.0; IBM Corp., Chicago, IL, USA). The participants’ sociodemographic characteristics were analyzed by the frequency, percentage, mean, and standard deviation. The correlations between social beliefs, personal beliefs, and COVID-19 precautionary behaviors were analyzed using Pearson’s correlation coefficients. Model fitness was confirmed using the normed chi-square, Tucker–Lewis Index (TLI), comparative fit index (CFI), standardized root mean square residual (SRMR), and root mean square error of approximation (RMSEA). The mediating effect of personal beliefs in the relationship between social beliefs and COVID-19 precautionary behaviors was identified through a covariance structure analysis using the maximum likelihood method, and the statistical significance was determined through bootstrapping. The significance level was set at *p* < 0.05.

## 3. Results

### 3.1. General Characteristics and Descriptive Statistics of Participants

In this study cohort, the mean age was 21.73 ± 4.50 years; there were 154 women (77.0%), and the participants obtained COVID-19-related information mostly through the Internet (67.0%). A total of 97 participants (48.5%) had experienced involuntary social isolation owing to COVID-19 (Table 1). Among social beliefs, the mean scores for social cynicism and social complexity were 2.70 ± 0.73 and 4.21 ± 0.59 (out of a possible 5 points), respectively. Among personal beliefs, the mean scores of the perceived benefits, perceived severity, perceived susceptibility, and perceived barriers were 4.24 ± 0.69, 3.16 ± 0.94, 3.74 ± 0.76, and 3.08 ± 1.05 (out of a possible 5 points), respectively. The mean score of COVID-19 precautionary behaviors was 3.69 ± 0.74 (out of a possible 5 points).

### 3.2. Correlational and Descriptive Statistics

Perceived benefits and perceived barriers were positively correlated with social cynicism (r = 0.22, *p* = 0.002 and r = 0.31, *p* < 0.001, respectively); the perceived severity and perceived susceptibility were positively correlated with social complexity (*r* = 0.22, *p* = 0.002 and *r* = 0.28, *p* < 0.001, respectively). COVID-19 precautionary behaviors were positively correlated with perceived benefits (*r* = 0.34, *p* < 0.001), the perceived severity (*r* = 0.24, *p* < 0.001), and the perceived susceptibility (*r* = 0.25, *p* < 0.001) but not with social cynicism (*r* = 0.11, *p* = 0.131), social complexity (*r* = 0.05, *p* = 0.482), and perceived barriers (*r* = 0.11, *p* = 0.123; Table 2).

### 3.3. Fitness of the Hypothetical Model and Path Analysis

The fitness of the model was calculated to be χ^2^/*df* = 1.64 (*p* < 0.001; TLI = 0.92, CFI = 0.93, SRMR = 0.06, and RMSEA = 0.06). The model was suitable because TLI and CFI were ≥0.92, and SRMR and RMSEA were ≤0.08 [32]. Social cynicism influenced the perceived severity (β = 0.26, *p* = 0.012) and perceived barriers (β = −0.31, *p* = 0.002); social complexity influenced the perceived susceptibility (β = 0.34, *p* = 0.002) and perceived benefits (β = 0.36, *p* = 0.002); perceived benefits (β = 0.41, *p* = 0.003) directly influenced COVID-19 precautionary behaviors; and social complexity (β = 0.94, *p* < 0.001) significantly and indirectly influenced COVID-19 precautionary behaviors. However, the perceived severity (β = 0.16, *p* = 0.210), perceived barriers (β = 0.01, *p* = 0.953), and perceived susceptibility (β = 0.15, *p* = 0.210) did not directly influence COVID-19 precautionary behaviors (Table 3). The results of hypothesis testing are shown in Figure 2.

## 4. Discussion

This study examined how social and health beliefs influence the COVID-19 precautionary behaviors of college students. A significant finding was that social complexity was related to COVID-19 precautionary behaviors and was mediated by perceived benefits. Therefore, prompt information on COVID-19, the government’s immediate response to the situation, and a positive perception of social complexity, such as policy flexibility, will further improve infection prevention behaviors by elucidating their effectiveness.

Social cynicism was correlated with the perceived severity and perceived barriers among personal beliefs. As social cynicism is based on a negative view of human nature, individuals with such a worldview exhibit worry, anxiety, withdrawal, and a defensive attitude [32], supporting the finding that social cynicism was correlated with the perceived severity and perceived barriers. Social complexity was correlated with the perceived susceptibility and perceived benefits among personal beliefs, which reflects the belief in complexity and variability of how various social problems are solved and outcomes are achieved [33]. Thus, social complexity is associated with a preference for variety and novelty, as well as with intellectual curiosity [32], which supports the findings regarding its correlations with the perceived benefits and perceived susceptibility. Accordingly, social beliefs are related to individuals’ health beliefs, although the association varies depending on the factors involved. Social beliefs originate differently by age, culture, and group [34,35]. Therefore, it was determined that different interventions for personal health beliefs should be applied to promote healthy behaviors and identify the social beliefs of college students.

COVID-19 precautionary behaviors of college students correlated only with perceived benefits among personal beliefs, whereas social complexity, among social beliefs, was correlated with COVID-19 precautionary behaviors, as mediated by perceived benefits. Direct comparison is difficult because studies using the same variables are rare, but the results of previous studies showing that perceived benefits are positively correlated with COVID-19 preventive behaviors [12], and that the social environment and perceived benefits are related to COVID-19 infection prevention behaviors [36], can be seen as similar. Individual behavior is influenced by the social environment, which can be examined in relation to social complexity. The social complexity referred to in this study is a concept that includes various policies, solutions, and actions according to the situation. Therefore, the rapid change in policies, guidelines, and quarantine measures according to the COVID-19 situation, or the rapid spread of such information, is adding to the social complexity of infection prevention. This is to improve the cognitive benefit of being able to prevent COVID-19 infection by complying with rapidly changing quarantine measures that are recognized as a beneficial environment for individuals. It can also be seen that these perceived benefits can enhance individuals’ infection prevention behavior.

These results suggest that it is necessary to promptly deliver and emphasize changes in the quarantine stage according to the situation and establish effective individual action guidelines. This includes information and policies on mask wearing, distancing, limiting gatherings, vaccination, and virus mutations that can reduce the likelihood of infection and reduce the severity of symptoms. As COVID-19 is highly contagious, preventive actions can benefit not only the health of each individual but also the entire community [12]. Therefore, it is also necessary to raise awareness of a “benefit for others” through health-related campaigns. Recently, there has been extensive research on COVID-19, and given the newness of the disease, a lot of new, sometimes contradictory, information is coming to light. In particular, college students are likely to encounter abundant and unclear information owing to the pervasiveness of social networking services and the Internet [20,37], which is expected to further increase social complexity. Therefore, efforts should be made so that necessary information can be rapidly disseminated through official channels (e.g., official school announcements, television news) in government agencies. It is necessary to enhance the perceived benefits through reliable information to improve COVID-19 preventive behavior among college students.

Based on the results of this study, the following suggestions can be offered. As social complexity increased, COVID-19 preventive behaviors increased through mediating perceived benefits. Therefore, first, it is necessary to quickly provide information about COVID-19 (e.g., incidence rate, transmission route, risk) so individuals can grasp the rapidly changing situation. Second, the government’s prompt response (including policy flexibility) is required according to the situation, and that response must be disclosed. Third, it is necessary to provide information on how beneficial COVID-19 infection prevention activities are to individuals (e.g., prevention of infection, reduction in the severity of symptoms after vaccination). Pandemics may occur frequently in the future. This study on COVID-19 infection prevention behavior can be used as basic data for determining how to change individual behavior in a future pandemic situation.

### Limitations

This study provides insight into the COVID-19 precautionary behaviors of college students. However, some limitations must be acknowledged. First, although this study was conducted online, most participants were female and college students of various majors took part; therefore, the results may vary depending on gender and the field of study. Second, the cross-sectional design precludes the determination of causal relationships. Third, because the survey used a self-reporting format, the participants may have given socially desirable responses, and thus, social desirability bias cannot be ruled out. Fourth, this study used a limited number of variables associated with social and personal beliefs. Therefore, it is necessary to consider research methods that control individual characteristics and to secure objective data, such as observational surveys. In addition, further research on various variables (e.g., self-efficacy, social support) that may affect COVID-19 precautionary behaviors should be conducted. Fifth, as only some Korean university students were included, caution should be exercised in generalizing the results to all students.

## 5. Conclusions

The findings of this study confirmed that consideration should be given to social complexity and perceived benefits regarding health beliefs to increase the compliance with COVID-19 prevention behaviors of Korean college students. As social beliefs differ by age, culture, and group, interventions for health beliefs should also be applied differently. In order to improve college students’ COVID-19 infection prevention behaviors, it is necessary to improve their positive awareness of social complexity, such as with prompt information on COVID-19, an immediate response to the situation, and policy flexibility, while emphasizing the effectiveness of such individual behaviors.

## Figures and Tables

**Figure 1 healthcare-11-01869-f001:**
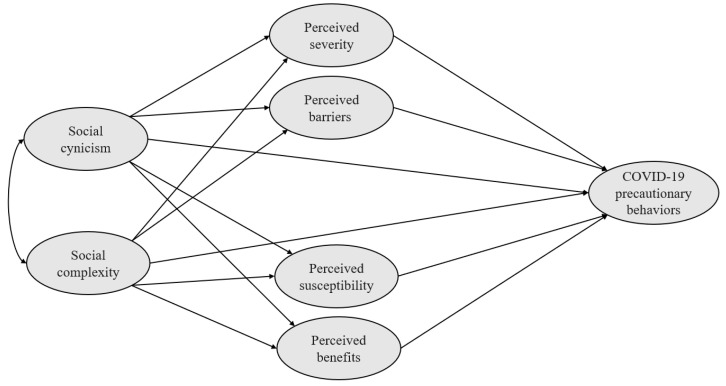
Hypothetical model of this study.

**Figure 2 healthcare-11-01869-f002:**
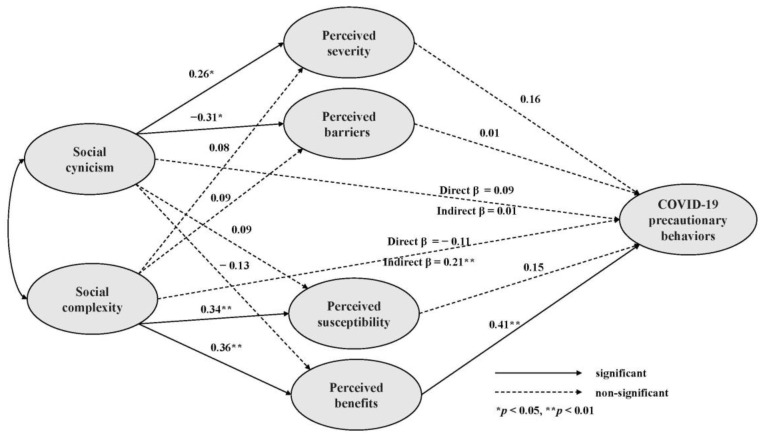
Path diagram of the research model.

**Table 1 healthcare-11-01869-t001:** Characteristics of participants and descriptive statistics of observed variables (*n* = 200).

Variable	Category	*n* (%)	Mean ± SD	Skewness	Kurtosis
Age (years)			21.73 ± 4.50		
Gender	Men	46 (23.0)			
	Women	154 (77.0)			
Grade	Freshmen	29 (14.5)			
	Sophomore	52 (26.0)			
	Junior	49 (24.5)			
	Senior	70 (35.0)			
COVID-19 information source	Internet	134 (67.0)			
	TV	22 (11.0)			
	SNS	44 (22.0)			
Isolation experience (COVID-19)	Yes	97 (48.5)			
	No	103 (51.5)			
Social beliefs	Social cynicism		2.70 ± 0.73	0.01	−0.28
	Social complexity		4.21 ± 0.59	−0.68	0.04
Personal beliefs	Perceived benefits		4.24 ± 0.69	−0.70	0.07
	Perceived severity		3.16 ± 0.94	−0.21	−0.47
	Perceived susceptibility		3.74 ± 0.76	−0.76	1.45
	Perceived barriers		3.08 ± 1.05	−0.10	−0.63
COVID-19 precautionary behaviors			3.69 ± 0.74	−0.11	−0.45

Note: SD = standard deviation; SNS = social network services; TV = television.

**Table 2 healthcare-11-01869-t002:** Correlations between variables and COVID-19 precautionary behaviors.

Variable		SCY	SCO	PBE	PSE	PSU	PBA	CPB
		*r* (*p*)	*r* (*p*)	*r* (*p*)				
Social beliefs	SCY	1						
	SCO	−0.09 (0.209)	1					
Personal beliefs	PBE	−0.09 (0.211)	0.22 (0.002)	1				
	PSE	0.22 (0.002)	0.07 (0.317)	0.11 (0.120)	1			
	PSU	0.04 (0.590)	0.28 (<0.001)	0.22 (0.002)	0.43 (<0.001)	1		
	PBA	0.31 (<0.001)	0.03 (0.659)	−0.02 (0.742)	0.50 (<0.001)	0.28 (<0.001)	1	
COVID-19 precautionary behaviors		0.11 (0.131)	0.05 (0.482)	0.34 (<0.001)	0.24 (0.001)	0.25 (<0.001)	0.11 (123)	1

Note: CPB = COVID-19 precautionary behaviors; PBA = perceived barriers; PBE = perceived benefits; PSE = perceived severity; PSU = perceived susceptibility; SCY = social cynicism; SCO = social complexity.

**Table 3 healthcare-11-01869-t003:** Verification of the hypothetical model.

Endogenous Variable	Exogenous Variable	SRW	SE	CR	*p*	Direct β (*p*)	Indirect β (*p*)
PSE	SCY	0.26	0.10	2.76	0.006	0.26 (0.012)	
	SCO	0.08	0.12	0.96	0.337	0.08 (410)	
PBA	SCY	−0.31	0.11	−3.55	<0.001	−0.31 (0.002)	
	SCO	0.09	0.13	1.14	0.256	0.09 (0.258)	
PSU	SCY	0.09	0.09	1.04	0.300	0.09 (0.358)	
	SCO	0.34	0.11	3.84	<0.001	0.34 (0.002)	
PBE	SCY	−0.13	0.06	−1.54	0.124	−0.13 (0.120)	
	SCO	0.36	0.07	4.36	<0.001	0.36 (0.002)	
CPB	PSE	0.16	0.08	1.38	0.168	0.16 (0.210)	
	PBA	0.01	0.06	0.05	0.964	0.01 (0.953)	
	PSU	0.15	0.09	1.39	0.164	0.15 (0.210)	
	PBE	0.41	0.11	4.22	<0.001	0.41 (0.003)	
	SCY	0.09	0.07	0.90	0.366	0.09 (0.443)	0.01 (0.938)
	SCO	−0.11	0.09	−1.15	0.249	−0.11 (0.280)	0.21 (0.003)
Goodness-of-fit statistics		χ^2^/df(p) = 1.64(<0.001), TLI = 0.92, CFI = 0.93, SRMR = 0.06, RMSEA = 0.06					

Note: CFI = comparative fit index; CPB = COVID-19 precautionary behaviors; CR = composite reliability; df = degrees of freedom; PBA = perceived barriers; PBE = perceived benefits; PSE = perceived severity; PSU = perceived susceptibility; RMSEA = root mean square error of approximation; SCY = social cynicism; SCO = social complexity; SE = standard error; SRMR = standardized root mean square residual; SRW = standard regression weights; TLI = Tucker–Lewis index.

## Data Availability

Data cannot be shared publicly because of restrictions by the Konyang University Institutional Review Board. Data are available from the Konyang University Institutional Data Access/Ethics Committee for researchers who meet the criteria for access to confidential data. Data requests can be addressed to the Konyang University Institutional Review Board (82-42-600-8466, kirb@konyang.ac.kr).

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
