# Peer review of "Social and Health Beliefs Related to College Students’ COVID-19 Preventive Behavior"

_healthcare, 2023, doi:10.3390/healthcare11131869_

Round 1

Reviewer 1 Report (New Reviewer)

Thank you for the opportunity to review this manuscript. The topic is interesting and presents a different approach to the pandemic that what not been consistently studied before.

In order to be suitable for publication I would like the authors to take into consideration the following comments:

- the Introduction section must insert information about statistical data regarding the country (when the pandemic started-ended and what where the rules imposed for college students), considering that the study was conducted after a considerable period of time after the return to school,

- data regarding college students worldwide and nationally,

The Discussion section must explain in congruency or in opposition the results of the study and to point the possible explanation for the specific results (hat could be due to national rules, cultural issues or personal behaviors).

I think that the study has a great potential to be publised.

English must be revised.

Author Response

Thank you again for reviewing the manuscript and for your comments.

Reviewer 2 Report (New Reviewer)

This paper explores the relationship between social beliefs and precautionary behavior against COVID-19. The researcher models the study in a way that allows social beliefs (specifically, social cynicism and social complexity) to directly influence behavior or indirectly be mediated by health beliefs. The researcher claims that only social complexity indirectly influences precautionary behavior against COVID-19 through perceived benefits.

This paper is simple yet has clear implications based on the study. The estimation model was tested before interpretation, making the results statistically valid.

However, I have the following major concerns:

  1. The researcher did not utilize all available data.

    • Table 1 shows that the researcher also collected information on sources of information and isolation experiences related to COVID-19. Although the number of observations is low, the researcher should at least mention the utilization of this data. For example, the researcher could report on the different social beliefs of individuals who experienced isolation compared to those who did not. Path analysis could even be used to examine different sources of information or isolation experiences. If the results are not significant, the researcher should still report them.
  2. The association between personal beliefs is not tested in the path analysis.

    • Table 2 shows some significant correlations between personal beliefs, such as PBA and PSU, with a correlation coefficient of 0.28 and p < 0.001. Therefore, the researcher might need to test these correlations in the path analysis or provide justification for why they do not need to be tested.

There are no minor concerns in this paper.

Author Response

Thank you again for reviewing the manuscript and for your comments.

Reviewer 3 Report (New Reviewer)

I would like to say thanks for the opportunity to review this article.

The article presented has an interesting theme with relevance for the improvement of health interventions in future pandemic situations.

Overall, the article has a scientific and appropriate writing, including all the components of a good scientific research. The title and abstract are related with the content. The keywords are linked to the research, and majorly are indexed.

In terms of writing, there are some acronyms and abbreviations that don’s have the full text in their first use (i.e. SNS, and others which the extended expression only appears after it’s use, ...).

Introduction allows the framing of the theme and the research itself. The main goal is appropriate. Methodology is scientifically appropriate, it’s not clear what sociodemographic questions/variables were included. Inclusion and exclusion criteria for the sample are redundant.

Results are adequate and allow to answer the goals. Some paragraphs are redundant with the tables in terms of information (i.e., 3.2). Discussion is done according to the results of the study, allowing a comparison and analysis with the scientific current evidence. Authors present limitations and strengths and valid conclusions.

References are pertinent, adequate, but being such an important and recent theme, it was important to include more recent references (almost 50% have more than 5 years and more than 30% have more than 10 years).

For this, we suggest the following corrections:

- include extended expressions in the first use of acronyms and abbreviations

- improve and accurate the inclusion and exclusion criteria of the sample

- include an overview of the sociodemographic questions in research instruments chapter

Thank you.

Author Response

Thank you again for reviewing the manuscript and for your comments.

Reviewer 4 Report (Previous Reviewer 6)

Thank you for the opportunity to review this interesting and important paper. Here are my comments:

ll. 33-36 requires a reference.

ll81-82 It is not clear what the author meant by "factors related to the perception of society". It is also not clear why "reward for application" is not included, since it is not about supernatural forces or religion, and reward for application seems highly relevant to this study. It is also not clear why social cynicism and complexity are related to COVID behaviors.

ll. 123-126 - I believe the exclusion criteria should not be the opposite of the inclusion criteria, but who in the included participants should be further excluded.

l. 132 - May I know how inaccuracies were determined?

ll. 136-140 - I believe this part should be under the data collection/procedures section. What do you think?

For Personal beliefs and COVID-19 precautionary behaviors questionnaire, can the author provide a sample item? This was provided for social beliefs.

There is a spelling error in line 187.

ll. 279-281 - It is not clear why the author stated that "However, if only the benefits are overly emphasized, then the effectiveness of the intervention may decrease owing to the strong optimistic tendencies of college students". This is not an objective of the study, and seems rather unrelated to this study.

It is also not clear to me, the explanation of why social complexity is associated with preventive behaviors, and mediated by benefits.

l. 272-273 "These findings support the results from a previous study wherein perceived benefits were positively correlated with COVID-19 precautionary behaviors" The author used the term "these findings" to cover findings on social complexity, perceived benefits, and preventive behaviors. However, this is not accurate. The cited study is only relevant for perceived benefits and precautionary behaviors. I suggest that the author be more precise here.

ll. 289-297 - I feel the explanation for this is not clear. It is not clear how income is a determining factor in perceived beliefs. It seems a bit far fetched to claim this in the discussion. It is also not clear why the author stated, "Nonetheless, excessive barriers can act as a factor that interferes with health behaviors" since this was not supported in this study. It seems to me that this paragraph is rather haphazard and not well-argued.

Overall, I would like to suggest that the discussion could be better written, with a better flow of logic and rationale without being too far-fetched.

In the limitations section, I wonder why a lack of homogeneity is an issue?

ll. 328-329 - Could the author provide some example variables?

Regarding the study implications, it is not clear how understanding the importance of social complexity on preventive behaviors could affect policy or interventions.

Author Response

Thank you again for reviewing the manuscript and for your comments.

Round 2

Reviewer 2 Report (New Reviewer)

Thank you for your response to my concerns. The paper quality has been improved and is sufficient for publication.

Author Response

I thank you and the reviewers for your thoughtful suggestions and insights. 

Reviewer 4 Report (Previous Reviewer 6)

Thank you for the opportunity to review this manuscript again. The paper is much improved.

I still don't understand why "reward for application" is not one of the study variables, considering it is a very important variable. If the author could add it in the limitations, and/or state the reason, that would be sufficient.

The edited parts of the paper need to be edited by a proofreader. For example, in the abstract Chi squared, the squared symbol needs to be superscripted. There are also other instances in which there are grammatical errors.

Author Response

I thank you and the reviewers for your thoughtful suggestions and insights. 

This manuscript is a resubmission of an earlier submission. The following is a list of the peer review reports and author responses from that submission.

Round 1

Reviewer 1 Report

The content is very up-to-date and addresses the restrictions and effects during the pandemic. The study is well collected. It is a pity that the results in the conclusions with regard to the aspects of the social, personal and societal level are not sufficiently differentiated. Here I recommend a revision.

Author Response

Thank you for your valuable comments. 

Reviewer 2 Report

Despite substantial work done by the authors, however, the poor quality of measurement instruments makes all subsequent work in vain.

Author Response

Thank you for your valuable comments. 

Reviewer 3 Report

Dear authors,

Thanks for the great opportunity for me to review the manuscript regarding the social and personal beliefs and their relationship with personal precautionary behaviors of COVID-19 infection. 

Introduction

1. In the Methods part, the authors provided a hypothetical framework. However, in the Introduction, the authors did not specify what they hypothesized. Specifically speaking, from the Introduction, the readers were not able to know that the authors postulated a direction that social belief would influence personal belief on the COVID-19 precautionalry behaviors. I would recommend the authors to incorporate the hypothetic framework and their hypotheses in the Introduction.

2. Since in the authors' proposed framework, social beliefs could potentially influence personal beliefs, it would be helpful if in the Introduction they could provide some evidence in the literature on this hypothesis. For example, evidence from other disease models.

Methods

1. How did the authors recruit the participants? Please specify.

2. Are there any inclusion and exclusion criteria of the participant recruitment?

3. Pg 3, line 111-112, "As this is a structural equation study, 111 the sample size needs to be 10 to 20 times the number of the observed variables." Can the authors provide a citation to back up this calculation?
4. The study used sequence equation modeling (SEM) method to evaluate the theoretical model. The authors should explicitly specify the method and include a brief introduction of SEM to familiarize the readers of this method.

5. The authors should specify the hypotheses and paths that they were testing using SEM.

6. The authors should specify the thresholds they used to assess goodness-of-fit of their models.

Results

1. The authors should comment on the fitness of the model.

Discussion

1. The authors can elaborate more on the significant findings, how do they explain the finidngs and what current literature can help support this finding. They should also explain the plausible mechanisms.

2. What are some potential concrete implications from this study?

Author Response

Thank you for your valuable comments. 

Reviewer 4 Report

It is important to increase the conclusions

Author Response

Thank you for your valuable comments. 

Reviewer 5 Report

I was very interested in the views that examined the relationship between college students' social and health beliefs and COVID-19 precautions. However, I was concerned about the following points. Please consider correcting them.

The definition of college students in this study is unclear. I believe that there will be a difference in views between medical college students and non-medical college students (e.g., law, economics, etc.). Please clarify this point.

Related to the above point, I would like to add to the limitations of the study regarding subject selection bias. Please clarify what can be inferred from the methods and results.

I am concerned that "COVID-19" and "SARS-CoV-2" are mixed in the sentence.

Since this survey is not a basic study, I do not believe that there is any discrepancy in the description regarding "COVID-19" alone.

Describe the information in a way that will enhance the understanding of readers with diverse backgrounds.

Author Response

Thank you for your valuable comments. 

Reviewer 6 Report

Thank you for the opportunity to review this manuscript. I have a few comments for the author's consideration.

Abstract. The abstract can be improved. Since this is a study that focuses on both social beliefs and personal beliefs, it does not seem suitable to state "Personal beliefs are crucial in preventing the coronavirus disease (COVID-19)" as the first sentence. I detect two sentences which are very similar, as follows: " Social complexity was 18 found to be related to college students' COVID-19 precautionary behaviors and was mediated by 19 perceived benefits." and later "Social complexity, as perceived by college students, was related to the COVID- 20 19 precautionary behaviors through mediating health beliefs (perceived benefits)." It does not seem reasonable to repeat this twice.

Introduction: The introduction needs to be extensively rewritten. The author uses the HBM, but apparently social beliefs is another theory/component apart from HBM. The organization of the introduction can be better organized. For example, the terms personal beliefs and health beliefs are used interchangeably. Ideally, the author should stick to just one term, and use it consistently throughout. 

The variable of social belief was also rather broadly defined. First, the author talked about "five belief factors of universal and general social axioms about individuals, the social and physical environments, and the spiritual world", and then the author provided definitions of the five components of social beliefs. How are these related the study? There is also no rationale given to substantiate why, out of the five social beliefs, the author only chose social cynicism and social complexity. In my opinion, reward for application is an important component to be considered in the study, but is not applied. Overall, I feel that the conceptual and theoretical framework are rather "haphazard" (sorry for using this word, but I can't think of another word at this moment), and there is inadequate justification for designing the study as such.

Under research instruments, the author stated, "Generalized beliefs about people, social groups, social systems, physical environments, spiritual world, and social phenomena that were proposed by Leung et al. [12] were used to measure social beliefs. The study focused on social cynicism related to prejudice against social groups and distrust of social institutions [16,17] as well as social complexity in constructing the questionnaire" Again, I'm not sure, is the author measuring people, social groups, social systems, physical environment, spiritual world, or the author is measuring social beliefs?

Methods: There should be more information given on how the data was collected, what online platform was used, how vast was the reach, because simply stating that 200 students from the ROC seems too general. Were they mainly from the university? What were their majors? Since the author was from a medical background, were the students also with a medical/clinical background? This would make a difference, and si a main weakness of the study.

A lot of the variables are not controlled in the SEM analysis. I think this is a problem. If the students were homogenous, such as from the same academic discipline, then it is ok to ignore these demographic backgrounds, considering social and personal beliefs can be influenced by these demographic factors. I feel the inability to control or the describe the academic background of the students is a major limitation of the study.

Results, Line 211 - it would be good to state the cutoffs that were used to determine goodness of fit.

Discussion, ll. 261-272 - it is not sure why perceived barrier is not significant in influencing precautionary behaviors among this study sample. Could it be Aligili's study targetted the general population, and this study was on students?

Thank you.

Author Response

Thank you for your valuable comments. 

Round 2

Reviewer 2 Report

The authors mentioned, "Social beliefs were assessed through social cynicism and social complexity factors in the tools presented by Leung et al. [12]."  Still, it's unclear how they use the tools; adopted the tool, or adapted the tool. If they followed the adoption or adaptation method for using other researchers' tools then where is the procedure in the article? This factor makes the whole research doubtful and shaky 

Author Response

Thank you for your comments. Edited and added content.

Reviewer 5 Report

Thank you for your careful revisions.

I have the impression that it is easier to read than last time.

I hope that it will be read by many readers and cited in the paper.

Author Response

Thank you for your comments. 

Reviewer 6 Report

Thank you very much for the opportunity to review this manuscript, and for the authors' efforts to improve it from the previous draft. I have a few recommendations:

The aim of the study was stated as "First, it confirms the correlation between social beliefs and health beliefs about college 130 students’' COVID-19 infection prevention behaviors. Second, it confirms the mediating 131 effect of health beliefs in the relationship between college students' social beliefs and 132 COVID-19 infection prevention behavior." Usually we do not write "confirms" in the aim of study, as this is usually used in the results or discussion section.

Author Response

Thank you for your comments. I modified the sentence according to your opinion.

  • Page 2-3: The purpose of this study is as follows: First, it identifies the correlation between social beliefs and health beliefs regarding college students’ COVID-19 infection prevention behaviors. Second, it identifies the mediating effect of health beliefs in the relationship between college students' social beliefs and COVID-19 infection prevention behavior.